# Adaptive Façades: Review of Designs, Performance Evaluation, and Control Systems

**Xi Zhang** [1,*] , **Hao Zhang** [2], **Yuyan Wang** [3] **and Xuepeng Shi** [4,*]

1 School of Architecture, Design and Planning, The University of Sydney, Darlington, NSW 2008, Australia
2 Department of Architecture & Civil Engineering, University of Bath, Claverton Down, Bath BA2 7AY, UK
3 School of Architecture, The University of Sheffield, Sheffield S10 2TN, UK
4 College of Architecture and Urban Planning, Qingdao University of Technology, Qingdao 266033, China
* Correspondence: zhang.sissi@outlook.com (X.Z.); shixuepeng@qut.edu.cn (X.S.)

**Abstract:** Adaptive building envelope systems can manage energy and mass transformation between indoor and outdoor environments, which contributes to the achievement of environmental benefits via reducing energy consumption and greenhouse gas emission while maintaining human comfort and well-being. However, the market penetration of adaptive façades (AFs) is far from sufficient, even though their capabilities have been recognized in research. Hence, this paper explores the factors hindering the growth of the market share of AFs, based on an exhaustive examination of designs, evaluation criteria and tools, and control systems. Insufficient commercial technology, inaccurate and incomplete performance data, and inconsistent evaluation criteria are demonstrated to be the factors that have hindered the widespread utilization of AFs thus far. Future research tendencies, including reducing costs, retrofitting existing building façades, developing building performance measurement tools, and building consensus evaluation criteria that favor the wide applicability of such façades in actual practice are identified.

**Keywords:** adaptive façade; evaluation; control

## 1. Introduction

Constant anthropogenic activities have contributed significantly to the early effects of global greenhouse gas (GHG) emissions, causing the world to experience climate change [1]. In particular, rapid urbanization and population growth have significantly increased global energy consumption, making GHG emission reduction necessary [2]. By 2035, worldwide energy consumption is projected to increase by 50% compared to the levels in previous decades. It is essential to reduce energy consumption from the source to address these serious problems. Buildings account for approximately one-third of both global energy consumption and GHG emissions [3]. Without transformation, the construction industry will likely be unable to restrain growing global energy demand. The European Union aims to achieve 32.5% energy efficiency by 2030 and a carbon-free building stock by 2050 [4]. To realize these goals, European legislation covers aspects ranging from building performance assessment and building automation to incentives (e.g., financing schemes) to assist building performance improvement and decarbonization. For instance, European COST Action TU1403 (Adaptive Façade Network (2014–2018)) encourages researchers to improve and create envelope technologies to control energy consumption [5], because building envelopes, as the interfaces affecting the energy exchange between indoors and outdoors, can be utilized effectively to improve the performance of buildings and achieve decarbonization. Walls, roofs, floors, and other components comprise building envelopes, and the wall is the main part of the envelope structure [6]. The International Energy Agency [7] has reported that huge energy savings can be realized by employing building envelopes, especially in energy-efficient building renovation, because many buildings will continue to be used after 2050.

To address energy performance challenges with regard to buildings, the role of building façades is being expanded from conservation to adaptation [8]. This expansion is occurring because adaptive façades (AFs), compared with traditional static façades, combine active and passive measures and can adapt to changing indoor and outdoor conditions in the short or long term by changing their shape [9] and color [10,11], among other aspects, to reduce the net energy consumption and ensure occupant comfort and well-being [8]. The current research on AFs is carried out from the following aspects.

Façades were first classified using the term 'AFs' in 2007 [12], but the term 'adaptive' has not been used uniformly across research thus far [13]. Based on Tabadkani et al. [14], AFs in this paper include the following typologies, 'active' [15], 'interactive' [16], 'responsive/media' [17], 'dynamic' [18], 'kinetic' [19], 'switchable' [10], 'smart' [20], 'intelligent' [21,22], and 'biomimetic' [23–25] (Figure 1); it is not possible to distinguish every category completely, as there is considerable overlap between each type. However, some studies, such as [26], show that AFs are abreast of these categories rather than included within them.

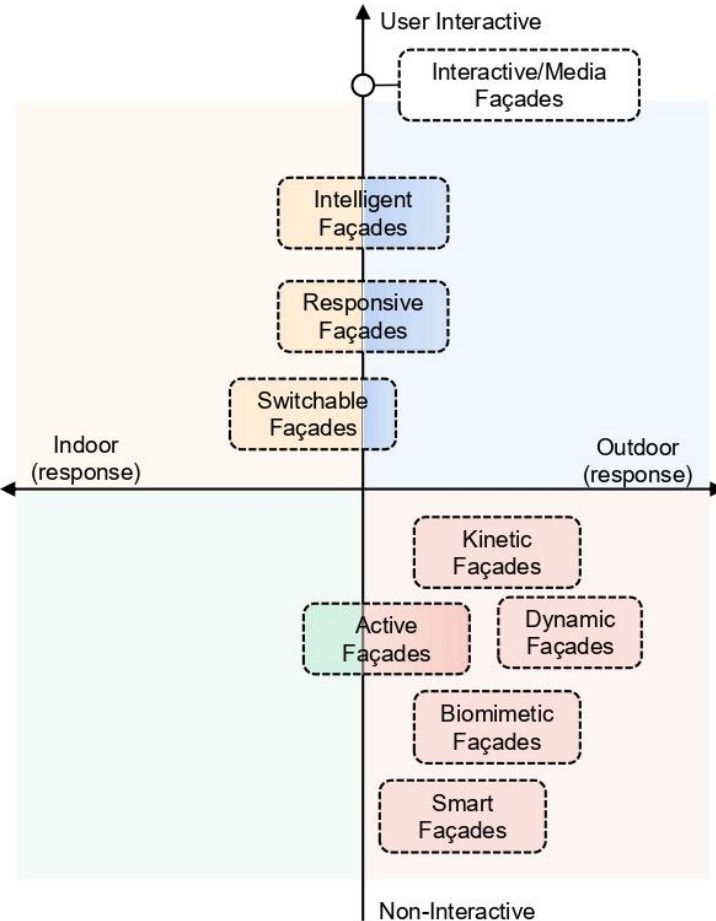

**Figure 1.** Main characteristics of adaptive façade typologies.

In terms of designs, based on functions, AFs can be classified as warm or cold façades. Based on structures, building adaptive skins can be identified as single or double façades. The concept of a warm skin is similar to that of a single skin, involving the use of insulation material on the inside or outside of the skin to improve the energy efficiency of the building. A cold skin is similar to a double skin, where a space is created between two layers of skin to allow air circulation and thereby ensure ventilation and cooling [27]. AFs can be designed according to the above basic principles. Then, combined with the needs of building users, AFs can be designed considering four aspects: indoor heating and cooling comfort, visual light comfort, natural air circulation, and energy consumption reduction.

Further, from a technical perspective, AFs can be divided into artificial technical and natural ecological skins.

Many techniques related to AFs can be acquired in the market [8,28], but the performance evaluation of such technologies has been limited [29]. The lack of consistent performance evaluation criteria seriously hinders AFs' widespread use and market penetration [29]. The salient feature of this type of façade is the variability of its responses to indoor and outdoor changes; however, many variables such as climate factors and occupant requirements exist, and integrating multiple variables for performance evaluation is challenging [25,30,31]. In addition, AFs can be created by changing shape and physical properties, but the performance evaluation of the combination of these features is still under investigation [24]. There are three main methods of performance measurement: digital simulation-based [32], experiment-based [30,33], and occupant survey-based [34] measurements. When new or renovated AFs are designed, the performance must be assessed by conducting building performance simulations (BPSs) during the design phase [35]. Compared with experimental methods, simulations are more widely available and are cheaper. However, the predicted values often differ considerably from the actual values due to the diversity of variables [28]. Further, the unique physiology, psychology, and roles of occupants result in considerable variations in survey results [36]. Therefore, performance evaluation needs to involve the entire AF life cycle and include all stakeholders [28] to drive market share gains.

When a device is designed to perform with responsive, superimposed control systems, it can be actuated to control indoor comfort and realize environmental benefits. The objective of employing AFs is to achieve a sustainable method of dynamically responding to the fluctuating external and internal environments in order to create a comfortable space for users [36] without affecting energy consumption, which is usually supported by a control system. The term 'degree of adaptability' was created to describe the complexity of AFs [37]. The degree of adaptability is determined by the number of possibilities; in other words, the more possible the AF state is, the more conducive it is to improving the indoor environment and comfort. The concept of module AFs has also been introduced, where an AF is composed of multiple modular units. Each unit has different states and can change them independently, and the diversity of unit quantity and states leads to diverse AF unit combination methods. Further, accurate modular unit control can improve AF adaptability.

The discussion in most relevant literature is based on one aspect, such as designs [6,13,14,20], performance evaluation [38–40], or control systems [41], or two aspects, such as designs and performance evaluation [25,42], designs and control systems [43], or control systems and performance evaluation [44]. Although researchers have realized that performance evaluation should be conducted throughout the life cycles of AFs and that all stakeholders should be considered [28], there is currently no clear and consistent evaluation standard to conduct performance evaluation and improve market share. Therefore, this paper is structured around three key aspects—designs, performance evaluation, and control systems—to explore and elucidate the factors hindering AF development via literature review and representative practical projects. Figure 2 shows the main structure of this paper.

The remainder of this paper is organized as follows. Section 2 describes the variable structures and materials, production properties, environmental benefits, and applications of AFs. Section 3 discusses the importance of performance evaluation, reasons for unsympathetic evaluation criteria, existing assessment criteria, and performance evaluation tools. Section 4 exhaustively explains the features of control systems, including influencing factors, methods, modes, strategies, and implementations. Section 5 addresses the ongoing challenges and trends. Finally, Section 6 concludes the paper by summarizing the key findings related to AFs.

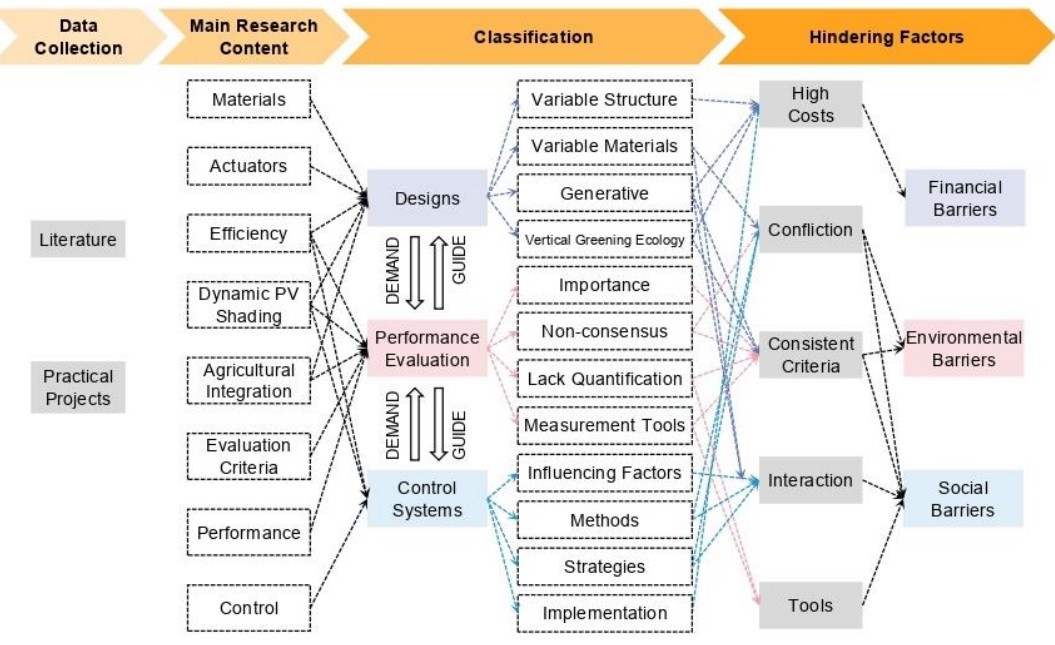

**Figure 2.** The main structure of this paper.

## 2. Designs

There are two main trends in AF design: artificial technology and natural ecology. Further, the directions can be divided into four types: innovation of the variable structure with parameter logic, development of variable materials for micro characteristics, utilization of building technology for resource production, and application of vertical greening ecology based on environmental benefits. In addition, to provide a more comprehensive and objective understanding of the development status and research frontiers related to AF design worldwide, practical projects, research prototypes, parts, and other related typical cases were sorted based on the four abovementioned aspects (Table S1). This study uses statistical data on actual AFs projects over the past 5 years (Figure 3). The result indicates that materials are used the most in practice.

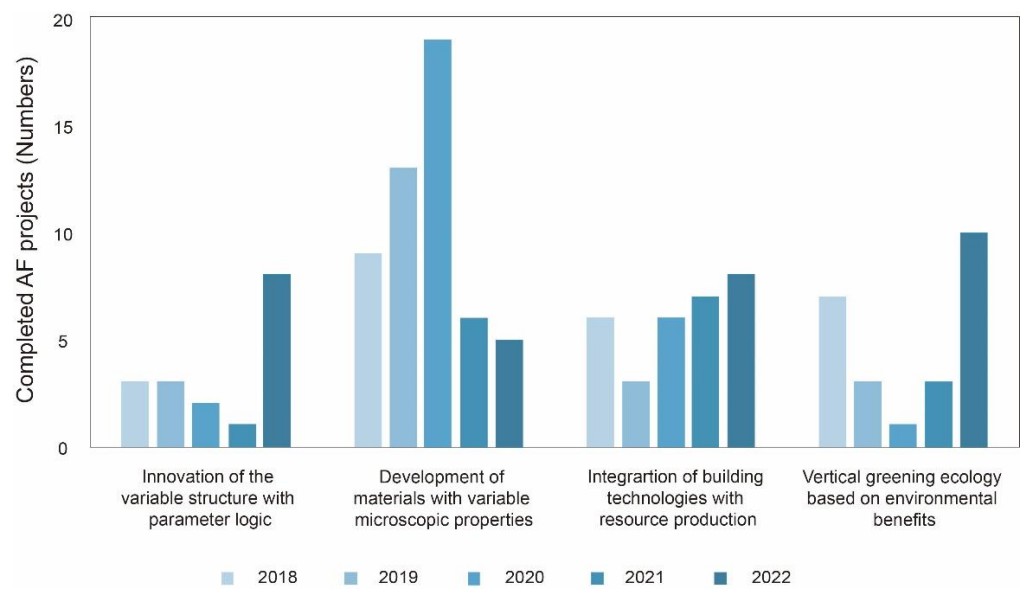

**Figure 3.** Practical AF projects.

*2.1. Innovation of the Variable Structure with Parameter Logic*

2.1.1. Outer Variable Shading System

As a common type of variable structure, an outer variable shading system has both practical and aesthetic properties as well as strong topological ability and applicability. The primary purpose of regulating solar radiation and indoor lighting is to prevent glare. It is characterized by dynamic responses to environmental changes to avoid the compromise of the traditional shading system in terms of the environmental regulation effect.

Sauerbruch and Hutton designed the Cologne Oval Office in 2010 [45]. The sunshade that rotates along the vertical axis in this building regulates light and solar radiation, while also serving as a façade element to fulfil the architectural aesthetic needs. Al Bahar Tower in Abu Dhabi, completed in 2012 [46], draws inspiration from natural forms, creatively using a hexagonal folding variable structure to adjust solar radiation and reduce indoor glare and programming to control the variable shading system to respond to the changes in solar radiation and incidence angle throughout the year. Thus, the solar radiation heat gain and building air conditioning demand are reduced by 50%. In 2016, Yazdani's CJ Blossom Park in Republic of Korea [47] also adopted the concept of folding variable shading, with retractable components to maximize daylight access.

In addition, many research institutions and laboratories have conducted relevant investigations. For instance, the Swiss Federal Institute of Technology in Zurich developed a dynamic photovoltaic (PV) sunshade system combining dynamic skin and PV cells and using a diamond matrix appearance and pneumatic control device. A complex algorithm control program was utilized to increase the light in the indoor environment, and considerable power output was achieved simultaneously [48]. Further, the Institute for Computational Design and Architecture and Institute for Architectural Structures and Structural Design at the University of Stuttgart investigated bionic-logic-based projects, such as responsive autonomous façade structures inspired by passive polyphasic plant motion and bio-inspired adaptive shading systems. They used the opening and closing principle of the plant *Helminthus* to design a variable shading system with a V-shaped cross-section by utilizing simulation software, which simplified the actuator and realized the opening and closing of the V-shaped shading component by a pair of pressure rods. In addition, Choi, Lee, and Jo [49] utilized variable element control of the outer layer as the entry point and developed three different response logics for a simple dynamic horizontal shading panel with the user as the center to satisfy the actual use needs of users, such as optimization of the indoor light environment, minimization of indoor energy consumption, and elimination of indoor glare.

2.1.2. Inner Variable Louver System

Compared with an outer variable shading system, an inner dynamic louver system is more controllable in terms of cost and is often integrated into a window system to form industrial production building parts. This approach is used in both renovated and new buildings. Double-layer blinds and a double-layer curtain wall with a controllable opening and closing cavity are included in this type of design. Because inner variable louver systems are mainly employed in the industrial field, they will not be detailed here.

*2.2. Development of Materials with Variable Microscopic Properties*

Compared with variable structures, variable material systems have the advantages of low complexity, low maintenance cost, and high operational reliability. At present, the design and development of AF based on variable materials mainly focus on phase change materials (PCMs) and programming materials.

2.2.1. PCMs

Researchers use the stability and energy storage properties of PCMs to improve the energy efficiency of buildings. Specifically, the heating and cooling loads are reduced by attenuating internal temperature fluctuations [50]. Currently, PCMs are usually combined

with walls to form sandwich wall structures [51]. In some cases, the position of the PCMs in the wall section is dynamically changed to utilize the PCMs to a greater extent [52,53]. PCMs are also combined with dynamic shading devices to improve indoor light environments and adjust indoor thermal environments simultaneously.

In recent years, researchers have also utilized combinations of PCMs and transparent windows [54]. The integration of windows with PCMs improves the utilization of solar energy, and the dual benefits of shading and energy buffering can improve indoor thermal comfort in summer and winter. Further, using PCMs in windows can affect visual performance. In 2018, the Delft University of Technology demonstrated a PCM-integrated window called Double Face, which consisted of PCM and glass layers that could regulate indoor temperature fluctuations and indoor light environments.

Often, such adjustment is not affected by the subjective opinions of the occupants, but instead, is a response of the inherent characteristics of the material under environmental stimulation. Therefore, to maximize the latent heat storage capacity of PCMs, it is next necessary to investigate how to use the material characteristic changes under the melting and solidification states to satisfy indoor environmental needs [55].

### 2.2.2. Smart Materials

A smart material is a synthetic material with one or more properties that can respond independently to changes in the external environment without the need for control by mechanical or electronic components. These environmental changes include stress, temperature, humidity, pH, magnetic field, light, and carbon dioxide concentration. At present, various smart materials are utilized in architecture research and application. The materials employed include shape memory alloys, shape memory polymers, thermo-bimetals, composite bilayers, electroactive polymers, wood, and hydrogels. Based on responsiveness, they can be divided into photosensitive materials, humidity-sensitive materials, and thermal materials. They can also be distinguished by production principle (according to the physical or chemical properties of the materials combined), anisotropy (the use of one material in programming processing to achieve characteristic responses in different directions), hierarchy, and angle of the multi-level structure (scale). From a manufacturing perspective, they can be designated as programming and non-programming materials. The applications of smart materials in adaptive skin can be divided into two types: skin coverings and actuators. Two types aim is to adjust incoming solar radiation and even generate electricity in transparent enclosed spaces: traditional nanomaterial technologies—electrochromic glasses and gasochromic windows—and emerging ones—nanocrystal in-glass composites windows, electrokinetic pixel windows [56], elastomer-deformation tunable windows, among others. Compared to other adaptive technologies, glass-level technology plays a pivotal role in existing building renovations, as it is generally easy to install.

### 2.3. Integration of Building Technologies with Resource Production
### 2.3.1. PV Integration

PVs are increasingly closely combined with construction façades, in multiple ways. One is PVs attached or applied to façades components—walls, glasses, blinds, among others—and the other is that PV replaces the traditional façade components and acts as a kind of façade component. Examples of the former include the PV Trombe wall, PV-wall, semi-transparent building-integrated PV glazing [57], and PV panels attached to blind slats, which use microscale (or even nanoscale) technologies with inconspicuous volume. Such adaptive-skin capacities include adjusting the cavity air flow rate, flow direction, and running mechanism to control energy consumption of buildings for space heating and cooling. Another example is a PV added-double curtain wall system, which is designed to provide electricity and heat simultaneously. In summer, cavity thermal pressure ventilation can be used for natural cooling of PV modules as well as ventilation and cooling of indoor spaces. At present, curtain wall-added PV modules can satisfy the

requirements of aesthetics and indoor vision, which proves the possibility of combining aesthetics, practicality, and technological innovation.

For the latter, PV as a sunshade component, usually called a PV shading device, constitutes an important part of PV integration. In recent years, considerable research on technology and methods has been conducted in Republic of Korea, China, Greece, and other countries, primarily focusing on office buildings. Zhang et al. [58] analyzed and sorted 24 kinds of existing PV shading prototypes, including horizontal plate, vertical plate, and frame-type systems, and believed that tropical areas should fully utilize solar resources to promote the research and application of PV shading technology. Dynamic solar shading is a further optimization of the above technology, which aims to fully employ the advantages of dynamic solar shading and PV power generation. Hofer et al. [59] verified the performance of modular dynamic PV shading with different arrangements, and the results showed that dynamic PV shading effectively improved building energy performance and residential comfort. However, methods of reducing the system cost and weighing the module spacing and occlusion influence remain topics for future research.

### 2.3.2. Agricultural Integration

A representation of building façade integration with agriculture is a vertical farming system.

Sustainable architecture has become a popular research topic [60]. It has the characteristics of vertical greening and the advantages of crop production, but differs from vertical greening in terms of the design method. The crop planting and picking processes require targeted design optimization. Kosorić et al. [61] conducted a questionnaire survey on the acceptability of building façade planting systems, with the results revealing high resident acceptance of such systems. Building skin planting has also gradually appeared in practical cases, and some service companies specialize in the design and promotion of exterior wall crop planting. In addition, different crop planting methods affect the design methods. Currently, the known planting methods include hydroponics, soil cultivation, and aeroponics [62].

### 2.4. Vertical Greening Ecology Based on Environmental Benefits

Vertical greening as a long-standing means of building environment regulation can be traced to the famous hanging gardens of Babylon in the Babylonian period. It is generally installed on the roof [63], but people have gradually noted its building façade applications. In the 1980s, Bartfelder and Kohler [64] conducted a preliminary study on the ecological function of building façade greening. With the gradual development of technology, new technologies began to be applied to vertical greening. Blanc investigated the first hydroponic vertical greening system in 1994 [65]. Since 2017, research has focused on the integrated use of vertical greening and PV solar energy systems in building exterior walls. The transpiration of plants can be used as a buffer layer between dynamic PV shading panels and building façades to provide cooling [66]. It has been experimentally proven that the temperature of a PV module is slightly lower than that of a PV module without a vertical green buffer, especially in summer.

### 3. Performance Evaluation

### 3.1. Importance of Performance Evaluation

Section 2 demonstrated that numerous innovative AF techniques have been developed, whose objectives are the improvement of energy efficiency and occupant comfort to some extent. However, in assessing these performance abilities, numerous scholars [28,67–69] have noted this task is challenging and essential, mainly due to the following two factors.

First, an AF, as a high-performance skin, is a complicated system. It generally influences multiple physical domains, including thermal properties, the amount of daylight, indoor air quality (IAQ), and energy [8,25,40]. These features may be correlated [40]; therefore, evaluating the performance of multiple domains simultaneously is essential. Further, unlike traditional static façades, high-performance façades are largely dependent on non-

linear changing behaviors, including climatic conditions, operational contexts, dynamic techniques [70], occupant behaviors [40], and maintenance issues [38]. Therefore, the ability of AFs to adapt to changing environmental conditions is nonlinear and cannot be fully characterized by traditional static performance indicators, such as the U-value, G-value, or T-vis [38]. Furthermore, constantly emerging dynamic and variable control features [70], such as interactions [69], are increasing the intricacy of building system management. With the well-being of occupants [71] being the ultimate goal of performance evaluation [34], indoor environmental quality (IEQ), energy efficiency, and occupant satisfaction need to be balanced. For instance, façades control sunlight penetration to provide sufficient daylight indoors to avoid dim and glare areas, and solar gain to regulate thermal comfort and minimize energy consumption.

Second, these advanced façades are constantly adapting to the ever-changing surroundings; hence, time-variation is inherent [40]. However, performance evaluation systems and frameworks for existing formal external façades, such as energy standards and regulations, have limited applicability to such advanced building façades [38,67]. Performance statements for AFs cannot be trusted without a standardized performance measurement protocol that can be consistently applied using performance criteria, although many scholars have demonstrated their advantages [38] in enhancing energy performance and improving environmental quality. Moreover, prescriptive specifications reduce potentially fatal mistakes in most cases and represent social needs [72], although they have the potential to hinder reformation and innovation. Currently, there are several ISO, EN, and national codes, most of which are used for project initiation [38] and continue to be used during the operational phases. However, there is no field standard for commissioning an entire façade system [38]. Commissioning standards for AFs that can be used throughout the building life cycle must be established to validate systems prior to operation and to adjust operation [73] to satisfy occupant requirements.

Overall, although it is challenging to quantify and evaluate AF performance [72], performance assessment is essential to further the development of such complex structural systems, to enable them to serve their practical role, and to increase their market share [28]. There is currently a lack of consensus regarding evaluation criteria categories to achieve a systematic performance assessment of AFs. Section 3.2 provides an analysis of the reasons for this phenomenon.

*3.2. Lack of Consensus Regarding Evaluation Criteria Categories*

As AFs are multi-functional, multi-disciplinary, multi-variate, and multi-standard (Figure 4), there is a lack of agreed-upon general evaluation indicators.

The first and most prominent feature of AFs is multi-functionality. Numerous studies [74–76] have shown that AFs have two significant functions: building energy management [77,78] and improving occupant comfort and health [11,79]. These functions are usually correlated with controlling indoor environment characteristics, such as solar heat, air flow, and noise. Researchers have attempted to superimpose the two aforementioned functions, but they are not easy to balance due to their competing requirements [69]. For instance, when indoor occupants want to maximize daylight, additional cooling is required to avoid overheating [80], which goes against the strategy of reducing building energy consumption. Individuals with potentially opposing expectations [81] in a shared space must also be considered. Hence, as human needs may exhibit transient changes, a balance between multiple functions considering different environmental stimuli must be realized to achieve adaptation. The aesthetic, economic, interactive, and durability functions [69,76,82] of AFs are also receiving attention, but require further in-depth research.

Second, AFs, as high-performance façades, involve the intersection of multiple disciplines, including physical, optical, and thermal simulation; materials science; chemical engineering; construction engineering; architectural design and building services engineering; and behavioral science [25,28,69]. Therefore, compared to traditional façades, their performance evaluation is more complex. Thus far, few studies have focused on commu-

nication across disciplines. Only Luna-Navarro et al. [69] have proposed a classification framework consisting of an interaction diagram and associated taxonomy notation to help transfer knowledge between disciplines, thereby facilitating the design of optimal solutions to achieve satisfactory occupant–façade interactions. More in-depth research on interdisciplinary information transformation would help establish multi-faceted AF evaluation systems considering design, construction, operation, and monitoring, and thereby, increase their market share.

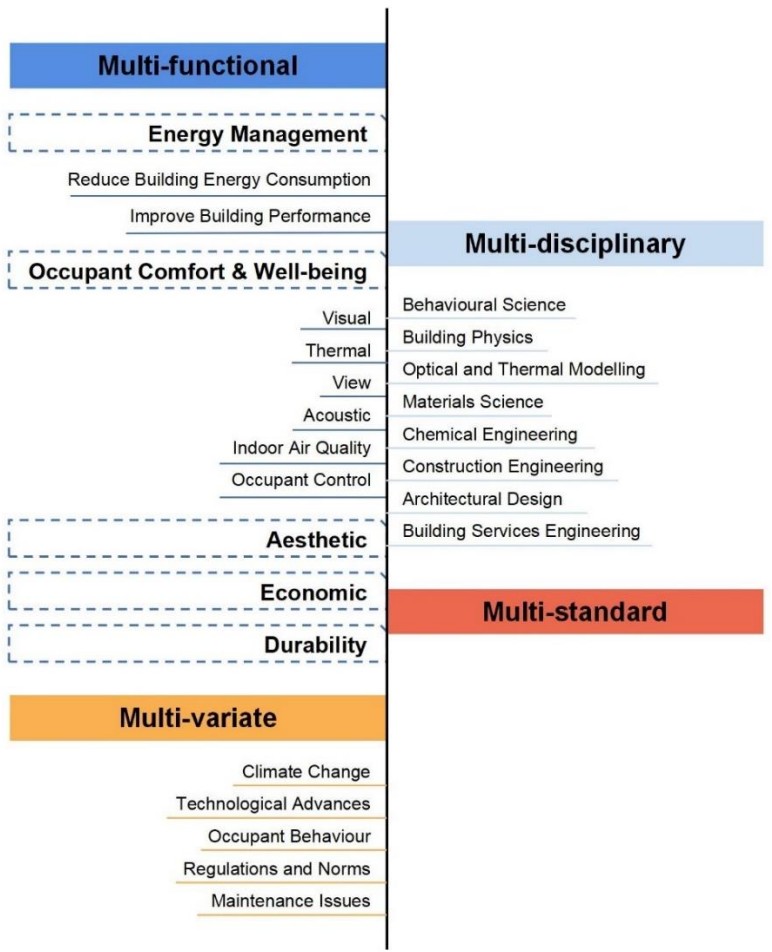

**Figure 4.** Distinctive features of AFs.

Third, multiple variables affect AF performance. Many of these variables, including climate change, technological progress, occupant behavior, regulations and norms, and maintenance issues [38,40,70], involve uncertainty because the response of such façades is nonlinear under dynamic environmental conditions, making performance evaluation highly uncertain. For example, when occupants use building equipment, such as heating, ventilating, and air conditioning (HVAC) systems and lighting and move and open doors and windows [83], the predicted energy consumption and actual performance deviate from the expectations. Thus, it is not appropriate to establish a general, standard, and static performance evaluation method. Further, it is not sufficient to evaluate the façade performance at a fixed point in time; instead, it is necessary to conduct life cycle assessments (LCAs) of the entire building, including during the operational phases, as well as post-occupancy measurements and monitoring [68].

Finally, AF performance evaluation is multi-standard, because of different evaluation criteria in different disciplines as well as changing environmental variables. Ferguson et al. [84] noted that multiple criteria must be considered when evaluating performance in different situations, as the design variables constantly change in tangible assessments.

For instance, adaptation to changing climatic conditions on a daily, seasonal, or yearly basis requires different assessment methods [68]. As another example, identifying independent standardized glare assessment indicators remains challenging when occupant conditions and local subjective responses differ [78]. In addition, when indoor personnel undertake special tasks, thermal, visual, and other comfort conditions must be changed, and standards other than the standard comfort model may be required.

### 3.3. Existing Performance Evaluation

AF performance evaluations focus on aspects affecting occupant comfort and well-being. As shown in Table 1, most studies on performance have mainly focused on the optical and thermal characteristics [8,34,85,86], especially those of dynamic shading façade systems. However, little is known about ventilation [86] or acoustic performance [34,82,87], even though academics agree that both functions cannot be easily ignored. The literature on ventilation often combines this aspect with indoor heat demand [6,20] (and, thus, is not listed separately in Table 1) but ignores its potential to improve IAQ.

**Table 1.** Aspects addressed in the literature on the performance evaluation of AFs.

| Refs. | Occupant Comfort and Well-Being | | | | | Environmental Performance | | | |
|---|---|---|---|---|---|---|---|---|---|
| | Optical (Dim and Glare Areas) | View | Thermal | Acoustic | Individual Control (Interaction and Requirements) | Energy Consumption | | Carbon Emissions | |
| | | | | | | Reduction | Generation | Reduction | Absorption |
| [88] | • | | • | | | • | | | |
| [89] | • | | | | | | | | |
| [33] | | | • | | | • | | | |
| [38] | | | • | | • | | | | |
| [72] | | • | • | | | | • | | |
| [77] | • | | • | | | • | • | • | |
| [90] | • | | • | | | • | • | | |
| [91] | • | | | | | • | | | |
| [92] | | | • | | | • | | | |
| [67] | | | • | | • | | | | |
| [68] | | | | | • | | | | |
| [93] | | | • | | | • | | | |
| [94] | • | | • | | | • | | | |
| [95] | | | | | | • | • | | |
| [28] | • | | • | | | • | | | |
| [86] | • | | • | | | | • | | |
| [34] | • | • | • | • | • | | | | |
| [96] | | | • | | • | | | | |
| [97] | | | • | | | • | | | |
| [98] | • | | | | | | | | |
| [82] | • | | • | • | • | • | • | | • |
| [44] | • | | • | | | • | | | |
| [78] | • | | | | | • | | | |
| [69] | | | | | • | | | | |
| [99] | | | | | | • | | | |
| [100] | | | • | | | • | | | |
| [101] | • | | • | | | • | | | |
| [36] | • | | • | | • | | | | |
| [79] | • | | • | | | | | | |
| [102] | • | | • | | | | | | |
| [76] | • | | • | | | • | | | |

In terms of improving occupant comfort and well-being, researchers have emphasized that individual control, including interaction and adaptation to individual requirements [34], is the main factor in façade evaluation. Due to the conflicting interests of operators and occupants [70], the former, who are concerned with the cost of energy usage and maintenance, often limit the control rights of the latter, who are concerned with satisfaction and experience. Fortunately, users have become aware of the connection be-

tween interaction and their own comfort and well-being. To realize interaction between the occupants and the façade, sensors or interactive devices are needed to sense user behaviors; further, the information must be processed by the control strategy and the outer wall actuators must be executed (the details of this process are provided in Section 4). If user-centered interactions are not considered, the resulting design decisions and control strategies may be unacceptable and unsatisfactory [34].

Interactions must also account for distinct individual requirements. Allowing personalization in AFs may result in higher occupant satisfaction and performance. Additional integrated building management systems (BMSs) can also monitor façade operation and help manage distinct individual requirements to establish a balance between façade operation execution and individual control. In order for BPSs not to hinder the market uptake of AFs [70], they may require further investigation.

Most research on environmental performance has focused on reducing energy consumption efficiency [77,88,101] as well as energy production capacity [72,86], which offset the net energy consumption of buildings. For instance, Jayathissa et al. [90] evaluated the energy performance of adaptive solar façades. Each unit in the considered system rotated individually to a position that maximized energy efficiency, balancing power generation and solar penetration control while reducing energy consumption for heating, cooling, and lighting.

Very few simulations and studies have involved carbon emission capability (reduction and absorption) evaluation, although experts have affirmed this capability in interviews [82]; this is probably because reducing energy consumption can be considered equivalent to reducing carbon emissions. However, in the case of adaptive building skins with plants, such as productive façades [103] and green façades or living walls [104], this fact cannot be ignored when evaluating the carbon emission capacities of buildings because plants can absorb carbon. In addition, vertical greenery systems have been shown to improve urban thermal environments to a certain extent [105]; therefore, the larger, city-scale environmental benefits of AFs require additional research.

Many scholars have helped improve the overall performance evaluation of AF systems. For example, Kasinalis et al. [88] developed a framework based on a genetic algorithm (GA) to analyze and quantify the energy and daylighting performance of seasonally adapted façades. Attia et al. [28] proposed an evaluation structure that includes key performance indicators, proposing an evaluation of the necessity, performance criteria, and technical qualitative characteristics of building AF systems. Battisti et al. [75] developed an LCA tool to assess the sustainability of the initial stages of AF technology. In addition, Yitmen et al. [106] proposed an analytical network process model to determine the evaluation criteria for AFs in complex commercial buildings. An adaptive evaluation framework for dynamic façades was proposed in [76], emphasizing the combination of top-down and bottom-up approaches to evaluate kinetic façades.

In general, the current performance evaluation criteria are inconsistent and lack quantification. The participation of all stakeholders involved in the design, operation, and monitoring stages is necessary to identify unanimous evaluation criteria, quantify such criteria, and ultimately promote their practical application. In addition, further accurate evaluation of both occupant comfort and well-being and environmental performance is necessary, because sometimes users are not allowed to engage with and use technologies [28] and environmental benefits are neglected.

### 3.4. Building Performance Measurement Tools

There are three types of performance evaluation tools for AFs: BPSs, experiments, and personnel surveys. The first two are often used to test the façade's ability to improve the indoor environment and realize environmental benefits, whereas personnel surveys often reflect the subjective perceptions of occupants regarding this mechanism based on social roles.

BPSs are widely utilized in building performance evaluation. Their ability to accurately quantify the advantages of buildings with AFs can facilitate the evaluation of the potential of such skins [40,107]. Modelling and simulations can help designers test, assess the interaction between the design and performance of, and effectively promote the development of new technologies to optimize their performance and facilitate their use in the construction market. These are in line with Bianco et al.'s [35] emphasis on the advantages of conducting BPSs to investigate alternative design performance in the design stage. However, as mentioned previously, the simulation of AFs may be more complicated than the performance prediction of traditional static façades. One of the main reasons is that the existing performance simulation tools, such as EnergyPlus, ESP-r, ICE, and IES VE, were mostly developed during a period when adaptability was not emphasized [32,40,108], and are limited in user-friendliness [40]. Further, the integration of multiple indoor and outdoor variables to control the adaptive capacities of façades is still in the research stage [24,25,30]. In addition, performance evaluation by combining morphological and physiological changes has not been explored [24]. When conducting multi-variate, multi-functional, and multi-scale evaluations, distinct degrees of simplified models may be required [109]. Despite these limitations, it still can be combined with other tools to help support design decision-making [110], such as ensemble multi-objective evolutionary algorithms and multi-criterion decision-making tools [76,110].

Approximately half of the literature has involved the examination of simulation methods alone [6], but there are often obvious gaps in field measurements. Typically, AFs do not perform as well as expected in actual applications [28]. Therefore, in addition to simulation-based studies, experiment-based methodologies can facilitate the performance evaluation of AFs, such as by using an experimental model of a parametric camshaft structure to design an adaptive shade for a home office [111]. Experimental methods can supplement the neglected factors that cause the apparent discrepancies between practical and simulated results to a certain extent, thereby improving the credibility of the performance value. Favoino et al. [30] built a prototype called the ACTive, RESponsive and Solar (ACTRESS) façade and measured its thermal storage performance during winter. The thermochromic glazing performance determined based on the photochromic optical properties measured from experiments and hysteresis phenomena optimized the evaluation of daylight availability and glare [112]. However, using either laboratory or field experiments to analyze examples of adaptive mechanisms is not sufficient [31], possibly because of high costs and apparent complexities involved in the experimental evaluation of building envelope performance.

BPSs can be less time consuming, and different options can be considered and evaluated during the design phase to select the most appropriate option or to improve the option until it is suitable. Meanwhile, experiments are beneficial for the modelling of complicated dynamic kinematics [76]. Few studies have involved combinations of these approaches, although scholars have clearly noted that it favors the mass production of AFs [6].

Many researchers have stated that simulations and experimental methods are sources of knowledge for scientists [25,113]. However, scholars are increasingly also focusing on investigating AF evaluation by individuals, including occupants and experts, to identify performance evaluation criteria. In façade projects, achieving occupant comfort, well-being, and interaction is important [28,68]. Attia et al. [28] interviewed specialized façade engineers, façade contractors, and architects to help them identify trends that would facilitate the development of a performance assessment framework. Attia et al. [34] surveyed 70 employees in an office building regarding happiness in an open-plan office with dynamic louvres. The survey results showed that most people were dissatisfied with and concerned about this project, indicating that user-centered surveys need to be incorporated into building design and that continuous follow-up needs to be performed during operation to enhance understanding among stakeholders. Direct investigations of building occupant comfort and well-being parameters may be more reliable than experiment-based studies [89]. However, to improve the validity of surveys, it is necessary not only to select

one time of the year for the survey, but also to conduct multiple surveys continuously to offset potential unidentified factors affecting the survey results.

Surveys of experts in architecture-related areas also help cross the boundaries between disciplines and utilize the accumulated experience and professional perspectives of experts to facilitate the development and application of AFs. A report on an interview survey of façade engineering experts [82] identified parameters other than IEQ parameters, such as the feedback mechanism and system learning ability, which can help overcome the obstacles related to AF performance evaluation and delivery. After obtaining interview data, it is also necessary to use statistics to analyze the data further to identify the development trends. Experts in other fields, such as consultants, managers, and designers, must also be considered.

## 4. Control Systems

AFs can be divided into traditional AFs and non-traditional AFs (Table 2). The former increases the cost to stakeholders and hinders applicability in practice, and it is difficult to evaluate the façade performance in the design stage in this case. There are significant limitations to energy consumption reduction and the management of multi-objective control with inconsistent human comfort requirements [44].

**Table 2.** Characteristics of AFs.

|  | Examples | Advantages | Disadvantages |
|---|---|---|---|
| Traditional AFs | Shutters or roller shutters |  | Costly<br>Limitations in terms of energy consumption reduction and responding to user needs |
| Non-traditional AFs | AF | Can respond to short-term changes in the surroundings and occupant preferences<br>High flexibility and intelligence | Low adaptability of control strategy technologies |

Non-traditional AFs are better than traditional AFs in terms of aspects such as the use of roller shutters or shutters, balance between energy consumption and occupant comfort, and responsiveness to the surrounding environment. AFs exhibit dynamic behaviors that account for system adaptability and can respond to short-term changes in the environment or visual comfort preferences of occupants. A dynamic PV shading system is a classic example of an AF. It is composed of multiple sunshade modules, which are controlled independently and have high degrees of personal control, especially in shared office spaces. However, the lack of adaptability of the control technology limits its development [114]. In addition, sufficient simulation tools that can interact with users are absent and cannot be integrated into the control process [115]. It is challenging to develop fast and reliable technologies to build simulation-based control programs, because the real-time sky state estimation provided by simulations is not very accurate. Non-traditional AFs, which involve fully automated control strategies, should enable occupants to veto automated operations [116].

### 4.1. Factors Affecting Control

A compromise between human comfort and energy consumption produces a reasonable solution, although no study has simultaneously investigated all five main objectives: thermal comfort, sunlight, vision, glare, lighting, and energy savings [44]. The influencing factors of control are mainly related to weather and user behavior (Table 3). Researchers have recommended minimizing the uncertainties of these factors to avoid deviation from the expected performance [117]. Optimization measures are utilized to improve the response time and ability to study the weather and to respond to the real-time variability of user behaviors by optimizing the machine learning algorithm.

**Table 3.** Influencing factors of control and solutions.

| Influencing Factors of Control | Solutions |
| --- | --- |
| Weather | Appropriate response time and learning ability |
| Occupant behavior | Use of a self-learning machine learning algorithm |

### 4.1.1. Weather

A control parameter is defined as a sensor output that activates the response of the control strategy [118]. Control parameters are time-varying and include building information such as location and façade orientation. Weather factors (Table 4) include external conditions (solar radiation, global horizontal radiation, air temperature, and wind velocity) and indoor conditions (temperature, humidity, air quality, and illumination).

**Table 4.** Weather control parameters.

| Parameter Type | List of Parameters |
| --- | --- |
| Outdoors | Solar radiation, global horizontal irradiance, air temperature, and wind velocity |
| Indoors | Temperature, humidity, air quality, and illuminance |

### 4.1.2. Occupant Behavior

Scholars have noted the absence of policies to map the preferences of individual users in shared work environments [119]. Saving energy largely depends on the degree of automation of the control system, and the system performance depends on occupant behavior or acceptance [115,120–125]; therefore, additional consideration should be given to the personal preferences and needs of users when optimizing control algorithms to achieve better system performance, and thereby realize greater energy savings [44]. Given that occupant behaviors considerably affect building energy consumption, integrating them into control logic is challenging [125].

Occupant behaviors have not been incorporated into self-control strategies due to non-physical variables: (i) psychological parameters (personal aesthetic preferences), (ii) physiological parameters (personal preferences for temperature and lighting), and (iii) behavior and social parameters (the ability to control the behavior of occupants in a shared environment), which cannot be measured with typical sensors.

Han et al. [125] focused on reinforcement learning, giving special attention to multi-agent systems. They reported that the current shortcomings are mainly reflected in the implementation of occupant integration (considering occupant preferences) and occupant mapping (information about presence and occupant behavior) [119].

### 4.2. AF Control Methods

This section reviews recent developments in occupant-centric control strategies [126]. As shown in Table 5, there are three main AF control methods: occupant interaction, automatic interaction, and occupant automation. AF occupant automation includes manual, occupant-oriented, and predictive control [36]. The difference between occupant interaction and automated interaction is whether there is occupant intervention. Occupant automation is the combination of occupant interaction and automatic interaction, which makes up for the shortcomings of occupant interaction (energy consumption) and automatic interaction (lack of consideration of occupant preferences) and introduces the concept of predictive control.

Specifically, occupant interaction involves a manual shutter (such as a rope that is pulled or a lever that is turned) and an electric shutter (such as a button that is pressed and held to move the shutter to the required position). Occupant interaction control methods can be employed to adjust shadows according to privacy requirements and visual preferences, which is very effective in daylight [127]. They also result in higher satisfaction

than automatic sunshades and manual shading. However, manual operation requires continuous occupant attention and cannot correspond to physical variables to achieve high thermal performance of buildings [128]. It may also overcome the difficulties associated with the variety of occupant preferences [129].

**Table 5.** AF control methods.

| AF control Methods | Subcategories | Advantages | Disadvantages |
|---|---|---|---|
| 1. Occupant interaction, occupant-centered (with occupant intervention) [36] | 1. Manual shutter 2. Electric shutter | Shadows can be adjusted according to preferences [127] and satisfaction is high. | Continuous occupant attention is required, the accuracy is low [128], and variations exist between individuals; therefore, this method is not universal [129]. |
| 2. Automatic interaction, that is, automatic control strategy without occupant intervention (includes open-loop and closed-loop) [37] | Occupant-oriented: same principle as the adaptive comfort model. Automatic control [36]: occupants are more likely to accept automatic sun shading and lighting control. | Control is easy, with continuous adjustment and a real-time system. | Visibility and thermal comfort are poor, long-term energy prediction is not performed, the environment cannot be customized [130], applicability and popularization are poor [119], and energy is not saved [131]. |
| 1 + 2. User automation, user-centric operating system (user interaction integrated into shading control) | Predictive control (automatic + user) (combination of software and hardware). | Users control façade components according to their preferences. | |

Automatic interaction strategies are easy to control. In addition, adjusting blinds to control glare can reduce the load of daylight and enable continuous adjustment. The added closed-loop feedback also enables automatic control strategies to work in real time. However, automatic control strategies have poor visibility and thermal comfort, do not enable long-term energy prediction, and do not allow environment customization [130]. The occupant cannot veto [116], resulting in poor applicability and popularization [119]. In addition, automatic control causes blind adjustment with improper design [36]. Further, it cannot save energy; complex lighting/shading control systems lead to dissatisfaction and capacity load [131].

The occupant automation approaches combine the advantages of the occupant interaction and automatic interaction strategies. Specifically, occupants can control façade components according to their preferences automatically or via artificial intelligence operation (autonomous operation/intelligent sunshade control systems). However, occupants show higher satisfaction with manual or occupant-oriented control systems. If the control device is easy to use, occupants prefer to learn how to use the device, resulting in energy savings [36].

*4.3. Control Modes*

The AF control strategy is equally important in the design and operation stages. It should focus on optimal annual control in the design phase and a fast control algorithm or short response time in the operation phase. In the design stage, the AF design and operating system should be evaluated by performing a BPS; however, the built-in modelling method cannot assess the dynamic thermal behavior or random user behavior over time or the interaction with the building façade [116]. This situation usually leads designers to make conservative decisions. Moreover, the sun shading systems and architectural design layouts in shared offices are not as personalized as those in residential buildings [116], as residents can send requests to back-end algorithms to adjust AFs in real time based on two control modes, namely, energy saving and visual comfort (Table 6). Personal decision making is also considered to have significant impacts on the comfort of residents and the use of building energy [116]. Therefore, a new algorithm was developed to enable users of shared office spaces to implement some personalization according to their visual comfort needs [116].

**Table 6.** AF control modes (adapted from ref. [116]).

| Personalized Control Mode | Type of User | Definition | | Drawbacks |
|---|---|---|---|---|
| Energy saving mode | | When the workstation has no occupants (on weekends or after working hours), the main purposes are to control the sunshade system automatically to respond to electric solar radiation as an outdoor sensor and to control the indoor temperature, thereby reducing the cooling demand. | | |
| Visual comfort mode [132] | Passive | User behavior is independent of the daylight environment. | | The overall energy consumption is high. |
| | Active | This approach is sensitive to visual and thermal comfort, and adaptive behavior improves comfort by preventing glare or increasing indoor daylight, interaction with lighting [133], shadows [134], windows [135], and HVAC setpoints [136]. | | The time spent in an uncomfortable state of glare and outdoor vision is long. |

### 4.4. Control Strategies

The optimal control strategy is related to the dynamic response time [43]. AF control strategies can mainly be divided into classic control, advanced control, intelligent control, other control, and hybrid control (Table 7).

**Table 7.** Classification, advantages, and disadvantages of control strategies.

| Control Strategy | Classification | | Advantages | Disadvantages | |
|---|---|---|---|---|---|
| Classic control | Rule-based method | | Simple, intuitive, cost-effective, quick response feedback | Control delay, energy inefficiency | No learning, solving incomplete data and control challenges, and handling an infinite number of possible variables |
| | Proportional integral derivative [137] | | | | Difficult, inefficient, and long test time |
| Advanced control | Adaptive control [138] | Gain scheduling of feedforward adaptive control based on prior knowledge and self-tuning control based on parameter estimation [138,139] | High applicability, fast response, dynamic change of parameters, good stability, and improved energy efficiency [138] | Appropriate design model is needed [138] | |
| | Optimal control [140] | | It determines the optimal control rules for the dynamic AF and can pursue the lowest possible energy cost to ensure the health of indoor conditions [140] | | |
| | Model predictive control (MPC) [141] | Data-driven MPC [142] | Cost-effective [140], energy-efficient [140], multiple variable control [143], improved steady-state response [144], upcoming control action prediction, transient response enhancement [140], and computational time reduction [145] | Cannot identify the system model accurately | Unable to deal with external interference and user behavior and difficulty finding the best solution for large buildings [142] Survey data from façades and buildings |
| | | Hybrid model based on the energy balance equation First-principles models | | | Rarely used because of calculation requirements [146] |
| | Feedforward/feedback [146] | | The combination of feedback and feedforward can improve the overall performance [146], because if the AF behavior deviates from the expectations, the front feedback cannot correct the input [147] | The AF output is fed back as control input, and the feedback is prone to error, because during interference, the feedback may deviate from the defined set value and have a response delay | |

**Table 7.** *Cont.*

| Control Strategy | Classification | Advantages | Disadvantages |
|---|---|---|---|
| Intelligent control | Robust controls [148] | Although disturbances and uncertainties affect the adaptive elevation, it is stable over a defined operating range [148] | |
| | Genetic algorithm | Global and non-derivative-based optimization | Large amount of computation, long processing time [149], and single objective |
| | Artificial neural network | Management of a numerous data and inputs [140], fast tracking speed, and quick operation | Long time and high complexity |
| | Fuzzy logic | High precision [140], fast tracking speed, and high efficiency | Long run time, high cost [150], limited input variables, lack of real-time response, lack of feedback [140], and massive calculation amount |
| Other control | Strong learning control | Reliable control and good performance | |
| | Multi-agent control | Ability to handle control and optimization of complex systems | Requires supervision |
| Hybrid control | | No need for long-time learning or accurate system simulation | |

### 4.4.1. Classic Control

Classic control has the advantages of simplicity, intuitiveness, cost-effectiveness, and a quick response feedback mechanism [140]. However, due to discontinuous adaptation, the control delay results in energy inefficiency [151]. Classic control approaches include rule-based methods [43] and proportional integral derivative (PID) techniques [137]. Rule-based methods are mainly temperature control [43], which may lead to low energy efficiency, because learning, solving incomplete data and control challenges, and handling an infinite number of possible variables are not considered in the design stage. PID techniques include proportional, integral, and differential controllers [43]. The proportional controller compares the feedback signal with the set value and produces an output proportional to the error. The integral controller eliminates the error by changing the error value to 0 over time. The differential controller increases the system response and reduces overshooting by reducing the correction coefficient. PID controller parameter settings that are not sufficiently accurate cause difficulty and inefficiency, which have no effect in the long run [138]. PID controllers have limited performance, require long test times, and are not suitable for nonlinear and complex systems [140].

### 4.4.2. Advanced Control

Advanced control is also known as hard control. It can be used during dynamic changes and disturbances to deal with the uncertainty of unknown models [43]. However, it has high maintenance costs, high energy consumption, and low efficiency, regardless of diversity. Advanced control methods include adaptive control, optimal control, model predictive control (MPC), feedforward/feedback control, and robust control.

MPC is cost effective [140], is energy efficient [140], can control multiple variables [143], improves the steady-state response [144], can predict upcoming control actions, enhances the transient response [140], and reduces the computational time [145]. MPC enables accurate and simple prediction and minimizes the primary energy costs [152], using output and input data to determine AF behavior. MPC is the most commonly used in AF control, can solve predictive control problems to satisfy dynamic and comfort constraints [153], and can predict the future state of the adaptive appearance. It takes the best control measures, including the objective function, prediction range, decision-making time, control variables, optimization algorithm, and feedback signal [138]. However, MPC does not recognize the correct system model [140].

### 4.4.3. Intelligent Control

Intelligent control is also called soft control. It learns from past cases to perform tasks without having to be programmed with specific rules. Such processes reduce the risk of incorrect operation, and multi-objective control procedures can be configured [119], which reduce failures and enable comprehensive monitoring during operation. It includes the following three aspects:

- GA: A GA is based on global and non-derivative optimization and is a wise choice when seeking a dynamic optimization objective, whether or not it contains mathematical ideas [140]. However, this approach results in a large amount of calculation and long processing time [149]; therefore, it is only suitable for a single target rather than for multi-target selection or dynamic calculation.
- Artificial neural network (ANN): An ANN is a machine learning tool that learns the relationships between the inputs and outputs to predict AF performance. It includes input, output, neuron, and hidden layers [154]. Due to its ability to manage large amounts of input data efficiently and perform fast tracking [140,150], it is very suitable for predictive models, nonlinear identification, and control, as well as for non-mathematical models. However, the huge data processing requirements of ANNs cause their training to be time consuming [140] when processing highly complex data, especially for large ANNs.
- Fuzzy logic (FL): FL is based on fuzzification, if–then rules, the inference mechanism, and defuzzification. It is similar to human reasoning and based on language models. Its features include high precision [140] and fast tracking speed [150], and it can reduce the tracking range through mathematical models and then use direct control algorithms [150] to perform a rapid operation for a local fine search. However, the long-term tracking process and massive amount of calculations result in considerable time consumption that prevents a real-time response [140]. Moreover, when dealing with more accurate models, it lacks feedback [140] on learning strategies. It also fails in terms of high cost and medium-to-high complexity [150].

### 4.4.4. Other Control

Other control methods include reinforcement learning and multi-agent control. Reinforcement learning control provides a reliable control technique for the optimization of active and passive heat accumulators in buildings, thereby achieving better performance. Although the latter can handle the control and optimization of complex systems, it requires supervision to guide and coordinate the subsystem operation.

### 4.4.5. Hybrid Control

Hybrid control is a combination of intelligent control and advanced/classic control. Hybrid control is stable, fast, and professional and can overcome the issue of the lack of solutions by a single façade control system by performing sufficient data training. It can combine MPC with reinforcement learning control and does not require long-time learning or accurate system simulation.

### 4.5. Control Implementation Methods

The control system runs based on several materials, movement modes, and mechanical components (Table 8). The materials belong to three generations. The first generation includes crystalline silicon, mainly monocrystalline silicon and polysilicon, and has the advantage of higher power generation efficiency than the second generation, although its shading tolerance is poor and its efficiency decreases sharply [155]. The second generation consists of thin films, which are not constrained by shape and can effectively resist the shadow effect and high temperatures. Due to their sufficient comprehensive performances, the first and second generations are widely used in AFs. Monocrystalline silicon and copper–indium–gallium–selenide (CIGS) thin films are the most popular. The third generation contains the emerging PV technology. Regarding movement modes, the rotation and

translation modes are most widely used. Among the relevant mechanical components, the control system and actuator require more attention. For the control system, first, the signal must be input into the system through modalities such as manual input, sensors, prior internal information, manual programming, and the Internet. This input signal drives the controller action. The controller serves as the interface between the input device and the actuator and is driven by calculation. It mainly involves internal (closed-loop) control and external (open-loop) control. The internal loop control does not intervene in occupant interactions or external information input and accepts feedback loops. By contrast, external loop control can intervene in occupant interactions and does not accept feedback loops [115]. Specifically, the controller directly provides outdoor climate information through external sensors for comparison with test points [156]. Internal loop control is faster, more accurate, achieves higher performance, and cheaper than external loop control; however, differences in the calibration factors could cause inaccurate decisions. In addition, external loop control realizes some improvements in flexibility and energy savings in calibration. Electrical drivers and pneumatic/hydraulic actuators are the two main types. An electrical driver provides accurate control and a fast response time, although it has a complicated structure and requires very professional individuals and environment for maintenance. Pneumatic/hydraulic actuators perform better in terms of cost, safety, air source cleanliness, durability, instantaneousness of the reaction, simplicity of construction, and power; thus, pneumatic/hydraulic actuators can be generalized and standardized. However, pneumatic/hydraulic actuators have the characteristics of noise, low stability, and complex purification processes.

**Table 8.** Control implementation methods.

| Implementation Method | Classification | Example | | Advantages | Disadvantages |
|---|---|---|---|---|---|
| Movement mode [157] | Movement of rigid elements | Rotation/translation | | Basic type and widely used | – |
| | Movement of deformable elements | – | | Rigid body, widely used in small-scale movements, particularly in movements over large surfaces | – |
| | Soft and flexible architectural elements | Linear elements: fibers, cords, or ropes<br>Flat elements: textiles (widely used), woven, knitted fabrics, and thin films (widely used) | | Permanent shape changes, ability to retain their overall formal consistency, light, flexible, adaptable to diverse movements (hanging, rolling, gathering, or pleating), and clear visual and space division | – |
| | Elastic architectural elements | Steel spring<br>Rubber shock absorber | | Back to original form after deformation without extra external force | Not enough literature on size, durability, or visual quality |
| | Pneumatic elements [158] | Expand<br>Shrink | | Low volume and stored in a small space after shrinking | Unable to oscillate elastically between expanded and shrunk forms |
| Mechanical components [157] | Connection | Independent component<br>Bearing<br>Hinge | | Five degrees of freedom, where the maximum degree of movement is artificially controlled through constraints | – |
| | Control system (software) [157] | Input signal [159] | Manual input | No need for a variety of control methods | Insufficient intelligence |
| | | | Sensor | The actuator control unit, with self-processing and self-driving functions, can directly respond to the environment by sharing information with other adjacent units through the 'Information Centre' without obtaining any input from other adjacent devices [158] and intelligent buildings | Inaccurate |

**Table 8.** *Cont.*

| Implementation Method | Classification | Example | | Advantages | Disadvantages |
|---|---|---|---|---|---|
| | | | Prior internal information | No need for sensors or detectors | Insufficient adaptivity |
| | | | Manual programming | Can conform to various conditions through modification by users or responsible officers according to the operation system | Insufficient intelligence |
| | | | Internet | Can acquire extra information such as climate data and other updates from the manufacturer | – |
| | | Controller (hardware tool) | Internal control (closed-loop/active) | Fast, few errors, high system performance, and low initial cost | No occupant intervention, differences in calibration factors can lead to inaccurate decisions, and no preventability |
| | | | External control (open-loop/direct/passive) | High flexibility, preventability, includes occupant intervention, and energy saving in calibration | No possibility for automatic correction, no feedback, low stability, long response time, and low accuracy |
| | Actuator/driver [160] | Pneumatic or hydraulic | | Low cost, safe, clean air source, durable, instant reaction, simple components, high power [161], standardization, and generalization | Noisy, low stability, and complex purification process |
| | | Electrical (e.g., Arduino) | | Accurate control and fast response time | Complex structure and maintenance must be performed by highly professional experts and site conditions |

## 5. Discussion

Although AFs have gained increasing attention over the past decades, they are still not popular in markets; this is because their operation performance does not match people's expectations. Figure 5 illustrates the predictable realization obstacles of AFs from three aspects.

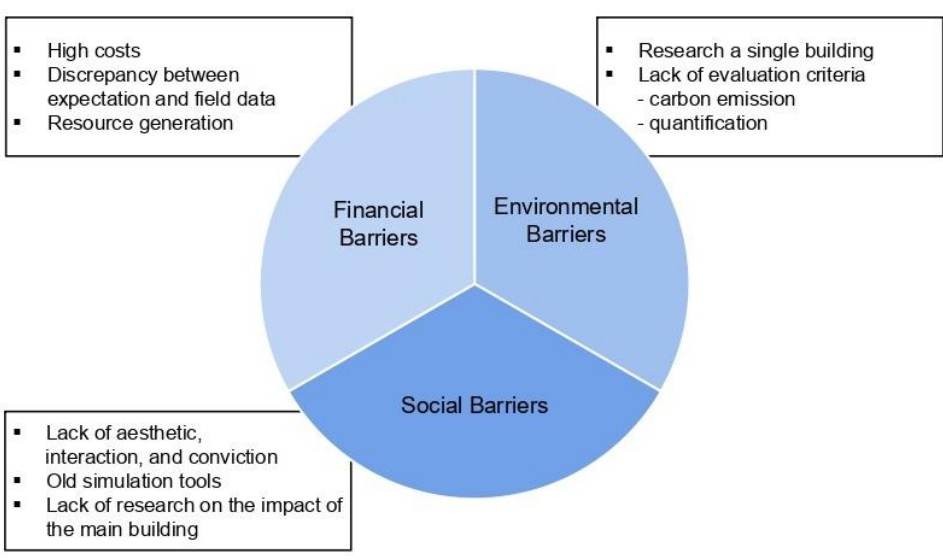

**Figure 5.** Barriers of AFs popularity.

First, financial barriers are considered as one of the most fundamental research factors. The current consensus is that the economic factors involved in building façade projects significantly influence the final decision-making processes. For example, capital expenses

and life-cycle costs may directly determine the final performance of AFs. As reflected by the AF projects mentioned in Section 2, the development of innovative technologies along with improvements in AF are likely to lead to high costs, a discrepancy between expectation and field data, and resource generation. Most projects currently use single-oriented technology, which is difficult to apply to other buildings. The lack of commercialization of such technologies is not conducive to increasing the market share of AFs. In addition, following the design step, Sections 2 and 3 illustrate that it is more fundamental and popular to use simulation methods as they are generally and readily available compared to field-experiment investigations. However, this approach is inadequate and inaccurate, as it includes different evaluation criteria that cause a gap between theory and practice. Improvement of the resource production feature can contribute to a reduction in energy consumption. If adaptive PV shading modules can overcome the situation in which the components are blocked from each other as well as outdoor shading, such as by the shadows of urban buildings and huge trees [59], it would further decrease the net energy usage of buildings.

Therefore, in the future, it will be necessary to strengthen the evaluation research on economic benefits to determine the optimal performance indicators and evaluation procedures [28] at the technical and control levels, as mentioned in the previous sections. This work will promote the large-scale standardized production of AFs and increase their market share while significantly reducing their cost. Moreover, although thermal comfort is often considered one of the most important evaluation criteria, control systems generally only focus only on optics. Future control systems must be developed to simultaneously study multiple occupant-related and energy-efficiency goals.

Second, limited studies are examining the environmental benefits of AFs. The existing research mainly focuses on the internal energy usage of a single room or building, rather than conducting large-scale research (e.g., at the community and city level). Examples include façades with dynamic photovoltaics that reflect additional sunlight onto poorly lit buildings or the sky to reduce the heat island effect, or the use of vertical farming to improve the microclimate in a community. Research specifically focused on the carbon emission of AFs is also limited, probably because reduction in the energy consumption of a whole building also leads to a reduction in carbon emission. However, if building façades were planted with plants, carbon emissions could be further reduced through a reduction in the carbon footprint of food transportation, cooling, and heating and the absorption of carbon dioxide.

Therefore, future studies must expand this research to realize environmental benefits at the community and city levels, and the entire life cycles of AFs should be covered while further quantifying the evaluation criteria for both indoor and outdoor environments. A combination of artificial intelligence tools or a subset of similar machine learning approaches [76] can be used to facilitate the development of façade assessments. Evaluation criteria can be established using bionic inspiration, combined with top-down and bottom-up methods, to perform complex adaptive evaluations.

Finally, dealing with social barriers would lead to a market share improvement in AFs. That is, high social acceptability indirectly causes popularity. Improvement of aesthetics, interactivity, and accuracy/predictability can be achieved via post-occupancy assessment, interactive platforms, and developed simulation tools. Attia's survey [67] of respondents who worked in AI Bahr Towers shows that about 50% of the participants did not hold positive attitudes towards thermal and lighting systems due to a lack of personal control and interaction with them. Therefore, enhancing the research and development of an interaction between AFs and occupants, such as linking sensors or incorporating BMSs, improves interaction ability and then social acceptability. Combined with developed performance, simulation tools [162] producing convincingly predictable data can also benefit society. For instance, when adding AFs to the façade of a completed building, it is necessary to not only to simulate their environmental benefits, but also verify the impact



of the additions on the main body, such as the main body's load-bearing structure and fireproofing and seismic proofing issues; however, there is a lack of research on this topic.

## 6. Conclusions

This study systematically presented the status quo of AFs in order to discover and identify the reasons hindering their popularity; it also suggested ways to improve the market penetration of AFs. This review contributes to stimulating AFs to achieve better energy performance in the future by removing the barriers preventing us from achieving environmental benefits; this can be achieved by significantly reducing energy consumption and greenhouse gas emissions while generally ensuring human comfort and society's well-being. Accordingly, AFs will be widely accepted and applied in society, thereby solving the social problems that humans are facing.

The single-project-oriented technology of AFs is definitely an efficient way to satisfy specific user requirements, but its high prices are unaffordable for all; this fundamentally leads to its unpopularity. Few studies have focused on the retrofitting of existing building skins, and little is known about the influences such appendages would have on the main building. Additionally, people may question whether AFs can actually ensure their comfort and well-being as has been publicized; this is because the performance of many developed façades may be tested only in a virtual environment, divorced from field testing, and simulations are often based on insufficiently advanced tools and inadequate criteria. Hence, simulation tools, field experiments, and personal control need to be considered in this context.

Moreover, AF capacities have not been sufficiently explored in line with improving the indoor and outdoor environments. Specifically, the overall criteria should be quantified, and AFs' outdoor benefits at community and city scales, such as reducing the urban island effect, should be clarified. In addition, there is a lack of research on the anti-disaster robustness of AFs; this might be because this is not the main function of building AFs. However, the establishment of consensus, normative, and well-established performance evaluation criteria are required.

**Supplementary Materials:** The following supporting information can be downloaded at: https://www.mdpi.com/article/10.3390/buildings12122112/s1, Table S1: Typical AF projects.

**Author Contributions:** Conceptualization, X.Z. and X.S.; Project Administration, X.Z. and X.S.; Supervision, X.S.; Visualization, X.Z., H.Z., Y.W. and X.S.; Writing—Original Draft Preparation, X.Z., H.Z. and Y.W.; Writing—Review and Editing, X.Z. and X.S. All authors have read and agreed to the published version of the manuscript.

**Funding:** This research received no external funding.

**Data Availability Statement:** Not applicable.

**Conflicts of Interest:** The authors declare no conflict of interest.

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
