# Peer review of "Adaptive Façades: Review of Designs, Performance Evaluation, and Control Systems"

_buildings, doi:10.3390/buildings12122112_

Round 1

Reviewer 1 Report

Please see the review in the attached file.

Reviewer 2 Report

  • In title of the article is” Design Approaches, Performance Evaluation, and Control Systems of Adaptive Façades (AFs): A Review” I can’t see any approach of design, and review of what? The title need improvement 
  • In the abstract section the author wrote “Adaptive building envelope systems play a significant role in achieving environmental benefits, including global energy consumption and greenhouse gas emission reduction, while also ensuring human comfort and well-being.” How? it is not clear, explain it  
  • In the introduction section the author wrote “Façades were first classified using the term ‘AFs’ in 2007 [12], but the term ‘adaptive’ has not been used uniformly across research thus far [13]. Various terms appear in the literature relating to façades, including ‘adaptive’ [5], ‘active’ [14], ‘interactive’ [15], ‘responsive’ [16], ‘dynamic’ [17], ‘kinetic’ [18], ‘switchable’ [10], ‘smart’ [19], ‘intelligent’ [20,21], ‘advanced’ [22], and ‘biomimetic’ [23–25]; this variety of terminology obstructs the widespread adoption and development of AFs [13].” This section is out of your paper scope, it is more general where reader need to understand your view and analyzing. 
  • What can help table 1 in your article.  But some of the table topics are distinct. 
  • Table 2 is overstated as topics, without a clear scope and not reflect any of design approaches 
  •  To keep track of all your topics and accomplish your goals, you must create a discussion section. 

Reviewer 3 Report

I consider the topic covered in the article to be of great interest. This information is very topical and linked to important aspects such as energy reduction and digitization. 

I propose improving the following aspects: 

-Line 43: “ And the wall is the main part of the envelope structure”. This is a statement that should be supported by some reference or quantitative data.

-The paragraphs formed by lines 54-59 and 60-71, are not connected. It is proposed to read the book xxx to relate the discourse

Reviewer 4 Report

Dear authors,

Thank you very much for the interesting research. The topic of the manuscript is in accordance with the Journal's requirements. The manuscript is very well structured and critical parts are explained. I appreciate the effort in writing comprehensive research on a very novel and interesting topic. The article is very long but concise.

The manuscript deals with the application of adaptive facades in structures. Particularly, it is a review paper on different aspects of adaptive facades. The topic of the paper is interesting for architects, structural and mechanical engineers and researchers in the field of façade systems, robotics, glass and polymeric materials.

I have several remarks.

  • Language: appropriate. Some mistakes are visible but nothing important.
  • Typos and wrong formatting: Some parts of the manuscript are missing. Please change it because a full review should be done when everything is included. For example, Table 3 is empty. Table 2 has formatting issues (one letter in the new row, empty spaces, etc.)
  • Line 37 – COST action TU1403 please cite the action and the STAR reports from the action. References 5 and 6 are not directly related to the action.
  • Table 1 – AF Network, COST Association – is it really an organization? It is an action
  • Do you have permission for all used figures?
  • Can you please have one paragraph on the influence of the AFs on load-bearing structure? Dynamic loads should be considered in different configurations. Also, please mention accidental load situations (fire, earthquake) and how they influence the usage of Afs. Is there a possibility to have optimized systems in earthquake-prone zones?

The manuscript is comprehensive and should be published. I hope that the above-mentioned suggestions can improve the quality of the manuscript.

Round 2

Reviewer 1 Report

The revision is correct and  satisfactory.

Reviewer 2 Report

The manuscript can be accepted